# Diameter-independent skyrmion Hall angle observed in chiral magnetic multilayers

Katharina Zeissler [1,2]*, Simone Finizio [3], Craig Barton [2], Alexandra J. Huxtable [1], Jamie Massey[1], Jörg Raabe [3], Alexandr V. Sadovnikov [4], Sergey A. Nikitov[4,5,6], Richard Brearton[7,8], Thorsten Hesjedal [7], Gerrit van der Laan [8], Mark C. Rosamond[9], Edmund H. Linfield [9], Gavin Burnell [1] & Christopher H. Marrows [1]

Magnetic skyrmions are topologically non-trivial nanoscale objects. Their topology, which originates in their chiral domain wall winding, governs their unique response to a motion-inducing force. When subjected to an electrical current, the chiral winding of the spin texture leads to a deflection of the skyrmion trajectory, characterised by an angle with respect to the applied force direction. This skyrmion Hall angle is predicted to be skyrmion diameter-dependent. In contrast, our experimental study finds that the skyrmion Hall angle is diameter-independent for skyrmions with diameters ranging from 35 to 825 nm. At an average velocity of $6 \pm 1\,\mathrm{ms}^{-1}$, the average skyrmion Hall angle was measured to be $9° \pm 2°$. In fact, the skyrmion dynamics is dominated by the local energy landscape such as materials defects and the local magnetic configuration.

[1] School of Physics and Astronomy, University of Leeds, Leeds LS2 9JT, UK. [2] National Physical Laboratory, Teddington TW11 0LW, UK. [3] Swiss Light Source, Paul Scherrer Institute, 5232 Villigen, Switzerland. [4] Laboratory "Magnetic Metamaterials", Saratov State University, 410012 Saratov, Russia. [5] Kotelnikov Institute of Radio-engineering and Electronics, Russian Academy of Sciences, 125009 Moscow, Russia. [6] Moscow Institute of Physics and Technology, 141700 Moscow, Russia. [7] Department of Physics, Clarendon Laboratory, University of Oxford, Oxford OX1 3PU, UK. [8] Magnetic Spectroscopy Group, Diamond Light Source, Harwell Campus, Didcot OX11 0DE, UK. [9] School of Electronic and Electrical Engineering, University of Leeds, Leeds LS2 9JT, UK. *email: k.zeissler@leeds.co.uk

In magnetic multilayer films the interface between an ultrathin magnetic material layer and a heavy metal layer gives rise to the Dzyaloshinskii-Moriya interaction (DMI) which favours chiral spin configurations. A consequence of the resulting competition between this interface contribution and the Heisenberg exchange interaction, which favours parallel spin configuration, is the stabilisation of particle-like magnetic domains called skyrmions. Skyrmions are topologically non-trivial magnetic textures with quasi particle-like properties[1–9] and the potential for low current density driven motion[10–13]. The combination of low current density motion with a predicted resilience to defects and edge roughness makes them potential candidates for future spintronic devices[14].

The non-trivial topology of the skyrmion has consequences for their response to a driving force. A skyrmion feels a Magnus force-like interaction when driven along a wire via a current-induced spin torque[15]. Under the influence of a driving force, a skyrmion will drift towards the wire edge, where the angle between the longitudinal and transverse motion is defined as the skyrmion Hall angle $\theta_{Sky}$[15]. In the work presented here, the skyrmion motion was induced through a driving force derived from electrical current pulses.

Assuming the rigid skyrmion approximation[16], and combining the Thiele equation[17] with the Landau-Lifshitz-Gilbert equation in a flat energy landscape, one finds that the skyrmion velocity components are given by

$$\tilde{v}_y = \frac{-\alpha_G \mathfrak{D} \mathcal{B}}{Q^2 + \alpha_G^2 \mathfrak{D}^2} j_{HM} \text{ and } \tilde{v}_x = \frac{Q \mathcal{B}}{Q^2 + \alpha_G^2 \mathfrak{D}^2} j_{HM} \qquad (1)$$

if a current is driven along the long axis of the wire (defined in this work as the y-axis). Here, $Q = \pm 1$ is the topological charge of a skyrmion. $\mathfrak{D}$ describes the dissipative forces acting on the skyrmion and $\mathcal{B}$ is linked to the spin Hall effect which converts the charge current in the heavy metal, $j_{HM}$, to the spin current that exerts spin-torques on the skyrmion, $\alpha_G$ is the Gilbert damping. The skyrmions experience a velocity component, $v_x$, towards the edge of the wire[9,18,19]. The skyrmion Hall angle is then given by $\tan \theta_{Sky} = -Q/\alpha_G \mathfrak{D}$. For skyrmions with a diameter, $d$, larger than the domain wall width, $\Delta$, at their edge, $\tan \theta_{Sky}$ can be approximated by[20]

$$\tan \theta_{Sky} \approx \frac{\pm 8\Delta}{\alpha_G \pi^2 d}. \qquad (2)$$

Assuming the domain wall width $\Delta = \sqrt{A/K_{eff}} = 4.6$ nm, based on typical values for the exchange stiffness, the effective perpendicular anisotropy constants, the Gilbert damping constant and the skyrmion diameter for multilayer systems, such as $A = 10$ pJm$^{-1}$, $K_{eff} = 0.47$ MJm$^{-3}$, $\alpha_G = 0.07$ and $d = 300$ nm, respectively, one expects a velocity independent $\theta_{Sky}$ of 10°. This expression depends only on the skyrmion geometry and the damping constant, and so does not predict any dependence of $\theta_{Sky}$ on the driving force. However, the above assumes a perfectly clean system. When defects are introduced deviations from the ideal case are predicted[21–24]. Experimentally, a current density, i.e., driving force dependence of the skyrmion Hall angle, was reported in multilayer systems[20,25,26]. In particular, a linear dependence of the skyrmion Hall angle on velocity was observed[20,25,26].

It has become more and more apparent that sputtered multilayer systems are prone to imperfections. While it is clear that nanoscale variations in the magnetic parameters of devices lead to skyrmion deformation and a wide range of stable diameters[27,28], the influence of defects on motion[29] and the skyrmion Hall angle remains an active field of debate[20,23,26,27,30]. On the one hand, skyrmion Hall angle deviations are attributed to dynamic deformation of the skyrmion during the motion[25] and on the other hand deviations are attributed to magnetic grains within the material[26,27,30] and defects[20–24]. Micromagnetic simulations have shown that the skyrmion velocity is dependent on the ratio between the magnetic grain size and the skyrmion diameter[30]. From this it is reasonable to suppose that there is an additional size dependence of the skyrmion Hall angle beyond the diameter dependence imposed by the topology. An experimental observation of the diameter dependence of the skyrmion Hall angle in the low velocity regime where pinning has a large influence on the skyrmion Hall angle is therefore of great importance.

Here, we investigate skyrmion motion through a 2 μm wide wire with diameters $d$ ranging from 35 to 825 nm. The skyrmions are stable at zero field and move up to 2 μm after the injection of 20 current pulses. As expected, the skyrmion Hall angle is non-zero. However, contrary to the standard theory embodied by Eq. 2, the skyrmion Hall angle was found to be diameter-independent. In our measurements, skyrmion-skyrmion repulsion and defects in the wire are dominating over the topology-driven skyrmion Hall effect.

## Results

**Imaging current-driven skyrmion motion.** Scanning transmission X-ray microscopy was used to observe the motion of skyrmions in 2-μm-wide wires fabricated from a Ta(3.2)/Pt(2.7)/[Pt (0.6)/CoB (0.8)/Ir (0.4)]$_{\times 5}$Pt (2.2) multilayer (thicknesses in nm) grown on an X-ray transparent Si$_3$N$_4$ membrane. Nucleation was achieved using current pulses with a current density of ~$5 \times 10^{12}$ Am$^{-2}$. This nucleation approach is a commonly used technique to nucleate skyrmions[31,32]. Superconducting quantum interference device-vibrating sample magnetometry (SQUID-VSM) (field applied in-plane) and polar magneto-optical-Kerr-effect (MOKE) magnetometry (field applied out-of-plane) show an easy axis out-of-plane (see Fig. 1a) with an effective anisotropy constant of $K_{eff} = 0.47 \pm 0.04$ MJm$^{-3}$. The magnetometry measurements were taken on a thin film sample sputtered onto Si in the same growth as the nanofabricated samples. Brillouin light scattering results from the thin film are shown in Fig. 1b. The observed frequency shift in the inelastically scattered light from the spin waves results in a calculated Dzyaloshinskii–Moriya Interaction (DMI) strength of $D = -1.1 \pm 0.1$ mJm$^{-2}$.

After the initial nucleation of skyrmions in the wire, 9-ns-long current pulses were injected with a maximum current density of $J = 1.2 \times 10^{12}$ Am$^{-2}$. Traces of the positive and negative current pulses are shown in Fig. 1c. The ferromagnetic layers, sputtered from an amorphous Co$_{68}$B$_{32}$ alloy target, support and sustain the motion of skyrmions at zero field (see Fig. 1d–o). Figure 1d–i shows single helicity scanning transmission X-ray microscopy (STXM) images after two consecutive current pulses. Figure 1j–o shows the skyrmion motion after the reversal of the current pulse direction (a new initial state was nucleated). The colour-coded arrows depict the direction of the positive (red) and negative (blue) current pulse. In both sequences one skyrmion is framed by a circle and its motion is tracked over 10 pulses. The skyrmions were found to move with the current direction, i.e., against the electron flow direction. An average skyrmion velocity of $8 \pm 3$ ms$^{-1}$ was observed in this image sequence. All these results shown in Fig. 1 were obtained under zero applied field.

Figure 2 shows the field-dependence of the skyrmion motion through another 2-μm-wide wire. The skyrmion area, $a$, was determined by counting the pixels within each skyrmion and an effective diameter was calculated using a perfectly cylindrical skyrmion model, i.e., the diameter is given by $d = 2\sqrt{a/\pi}$. Figure 2 a shows that the size distribution of displaced skyrmions tends to lower diameters as the negatively applied field is

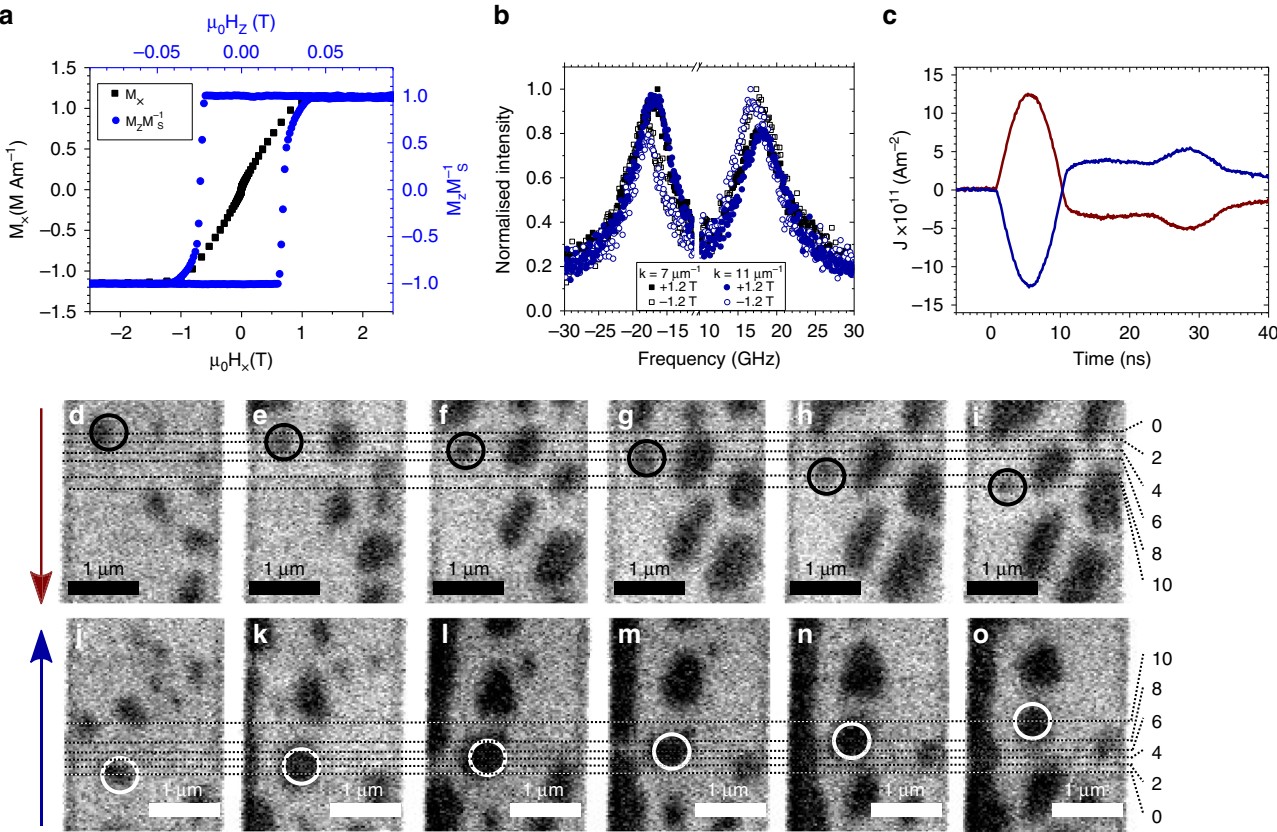

**Fig. 1 Zero field skyrmion motion.** Skyrmion motion was observed in a 2 μm wide Ta(3.2)/Pt(2.7)/[CoB (0.8)/Ir (0.4)/Pt (0.6)]×5/Pt (2.2) multilayer wire (thickness in nm) in 0 mT applied field. **a** Magnetisation, $M_x$, versus in-plane ($\mu_0 H_x$) and normalised magnetisation $M_z M_S^{-1}$ versus out-of-plane ($\mu_0 H_z$) field measured on a multilayer film using SQUID vibrating sample magnetometry and MOKE magnetometry. The saturation magnetisation, $M_S$ was measured to be $1.2 \pm 0.1$ MAm$^{-1}$. **b** Brillouin light scattering was used to measure the DMI strength, $D$, of a multilayer film. $D$ was extracted from the asymmetry of the Stokes and anti-Stokes lines due to inelastically scattering of the incident light from the propagating spin waves with magnon wavevector, and was found to be $-1.1 \pm 0.1$ mJm$^{-2}$. **c** Time-trace of a positive (red) and negative (blue) current pulse applied through the wire. **d–i** Single helicity STXM images after two consecutive 9-ns-long current pulses with current densities of $J = 1.2 \times 10^{12}$ Am$^{-2}$ applied along the direction of the red arrow (positive current) and **j–o** along the blue arrow (negative current). The dashed lines track the vertical position of the skyrmion centre throughout the pulse sequence. An average skyrmion velocity of $8 \pm 3$ ms$^{-1}$ was observed in this sequence of images.

increased. This is expected as the field polarity was chosen to apply a radial shrinking force on the skyrmions[28].

However, for applied fields greater than $-3$ mT, a shift towards larger average diameters is observed (see Fig. 2b). This is attributed to the observation that small skyrmions disappear after the first current pulse train. This is most likely to be a consequence of a combination of the static field, the Oersted field and the heating generated during the current pulse. A positive current pulse was applied from the top of the wire (along the $y$ direction as defined by the imposed coordinate system). The inset in Fig. 2b shows a measurement of the applied current pulse. The maximum current density of the 9 ns pulse used to move the skyrmions was $5.6 \times 10^{11}$ Am$^{-2}$. A static single helicity STXM image was taken after two consecutive current pulses separated by a delay of 2 μs. The centre of the displaced skyrmions was identified and tracked using the ImageJ TrackMate algorithm[33]. The skyrmion displacement and velocity was calculated from the change in the skyrmion centre coordinates identified by the algorithm before and after each current pulse pair. Figure 2c–l shows the centre of the moving skyrmions superimposed onto the initial state single helicity STXM image with respect to an out-of-plane field. Their motion was observed in 0, $-0.5$, $-1$, $-1.5$, $-2.0$, $-2.5$, $-3.0$, $-3.5$ and $-4.0$ mT fields (see supplementary information for videos of the images taken at all fields). As the field increases in strength, the length travelled by the skyrmion

decreases. This was seen to be due to an increase of the number of annihilation events. 16 skyrmions versus 21 skyrmions were nucleated in the case of exposure to a 0 and $-4$ mT field, respectively. After the first four current pulses the 0 mT data shows that all 16 skyrmions are intact, whereas, at $-4$ mT only 12 skyrmions remained.

**Skyrmion Hall angle**. Figure 3a shows the skyrmion Hall angle, $\theta_{Sky}$, evaluated from the displacement of the skyrmion centre position versus the diameter of the skyrmion prior to the displacement. The skyrmion centre coordinates of all images were identified using the TrackMate ImageJ algorithm. The raw diameters and corresponding skyrmion Hall angles were binned into 25 nm and 5° intervals, respectively. Figure 3b shows the average $\theta_{Sky}$ value within each diameter bin. The lines superimposed on Fig. 3b show the best fit employing a linear model (blue dash-dot-dotted line) and the expected behaviour using the rigid skyrmion model (grey dashed line, red line, and grey dotted line) with $\alpha_G$ taken to be 0.5, 0.07 and 0.02, respectively. The upper and lower limits were chosen as these are values typically used for similar multilayers in which skyrmion motion has been demonstrated[20,25]. A Gilbert damping constant of 0.07 was obtained by fitting the data using Eq. 2 and restricting the range of the fitted data to diameters of 175 nm and larger. Skyrmions

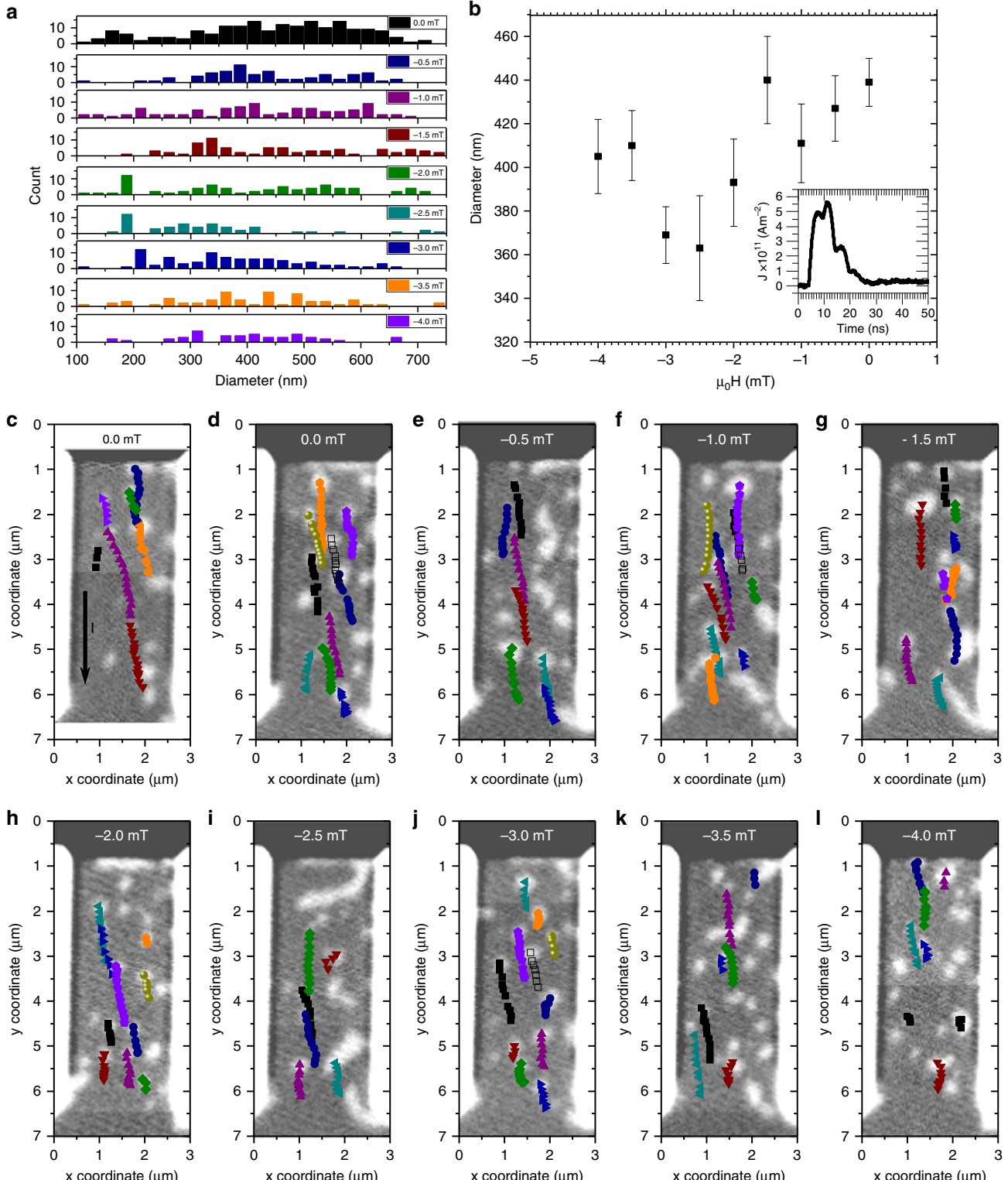

**Fig. 2 Current driven skyrmion motion.** Applied magnetic field-dependent current-pulse-driven skyrmion motion in 2-µm-wide Ta/Pt/[Pt/CoB/Ir]$_{\times 5}$/Pt multilayer wires. **a** Histogram of the skyrmion diameter distribution at each value of out-of-plane field. **b** Average diameter at each value of applied field. Standard error is plotted. As the field becomes more negative, the average diameter of the mobile skyrmions reduces until a critical field of −3 mT, at which point small skyrmions annihilate during the current pulse. The inset shows a trace of the current pulse applied from the top to the bottom of the wire. STXM images taken after two consecutive pulses separated by 2 µs capturing the skyrmion motion. **c–l** Initial state STXM single helicity images of the magnetic state in the wire (arrow in **c** indicates the applied current direction). Superimposed onto the images are the skyrmion centre positions after each two-pulse train (coloured symbols). The images were taken with in an out-of-plane field of 0, 0, −0.5, −1.0, −1.5, −2.0, −2.5, −3.0, −3.5, −3.5 and −4.0 mT, respectively. The peak current density was $5.6 \times 10^{11}$ Am$^{-2}$.

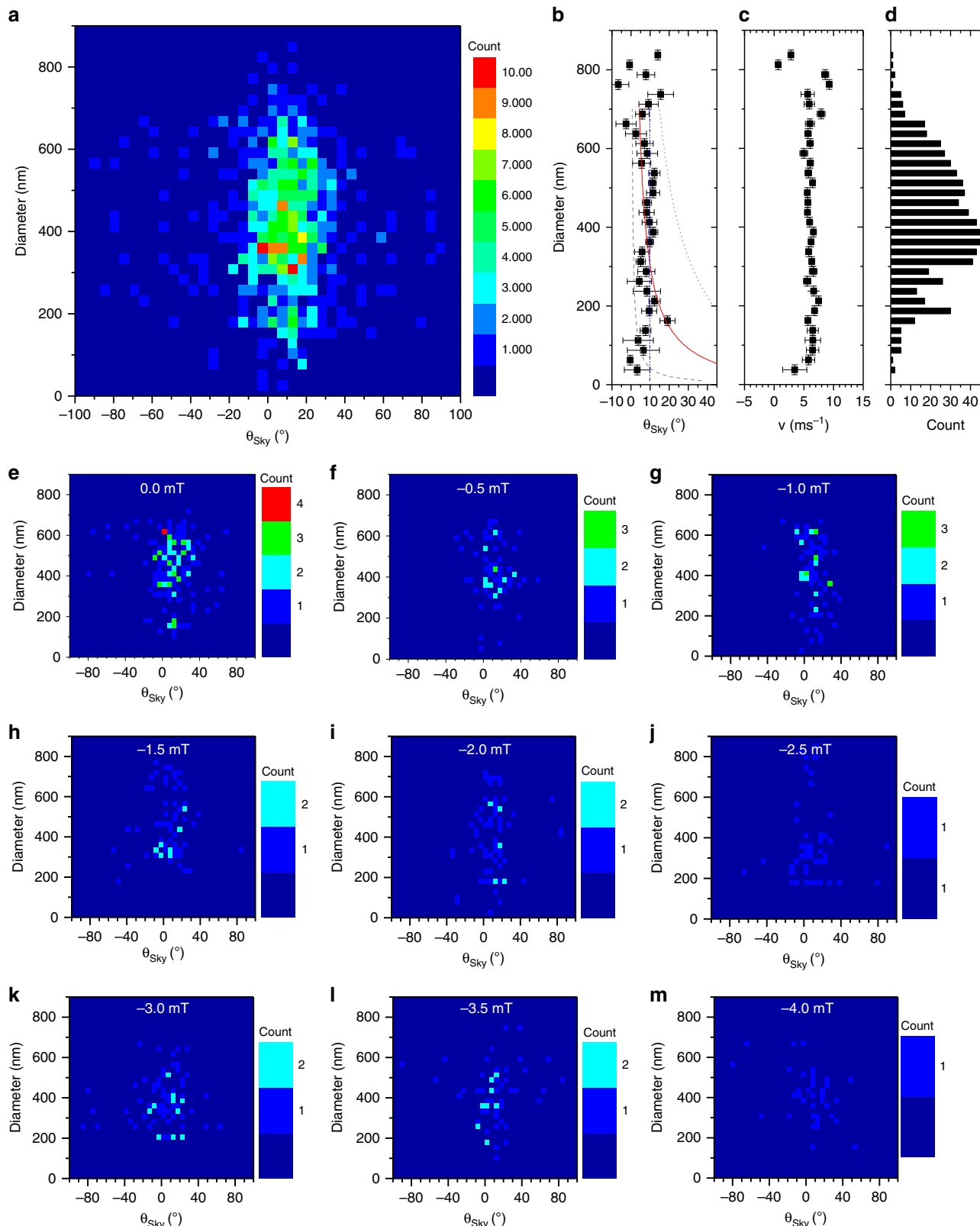

**Fig. 3 Diameter-dependence of the skyrmion Hall angle. a** Frequency count of skyrmion diameter versus skyrmion Hall angle $\theta_{Sky}$. The diameter was extracted by counting the pixel area of each moving skyrmion and evaluating the effective diameter for a perfectly circular skyrmion of the same area. The diameters and angles were binned into 25 nm and 5° intervals respectively. **b** Average $\theta_{Sky}$ evaluated from data in **a**, and **c** the average velocity versus diameter with the respective histogram **d**. Standard error is plotted. The diameter error is given by the image resolution. The blue dash-dot-dot line shows the best linear fit to the data, the grey dashed line, red solid line and grey dash dot line show the expected diameter dependence using the rigid skyrmion motion model with $\alpha_G = 0.5$, 0.07 and 0.02, respectively. **e**–**m** show diameter-skyrmion Hall angle data as a function of applied field. Adding all these plots together will give the overall plot in **a**.

with a diameter of 150 nm and smaller are observed in the initial, post nucleation state and hence are in regions with high defect density. An exchange constant of $A = 10$ pJm$^{-1}$ was assumed, and an effective anisotropy constant, evaluated from data in Fig. 1a, of $K_{eff} = 0.47$ MJm$^{-3}$ was used. However, contrary to the expectation from Eq. 2, no diameter-dependence was observed in the data. A constant average skyrmion Hall angle of $9° \pm 2°$ was found, as observed by the linear fit (blue dashed line). The skyrmion Hall angles below diameters of 150 nm do not follow the predicted trend of growing rapidly as the skyrmion diameter decreases.

Figure 3c shows the average velocity of the skyrmion within the same bins. The histogram in Fig. 3d shows the number of skyrmions within each bin. Figure 3e–m show the data displayed in Fig. 3a for different applied out-of-plane fields. The average velocity was found to be constant over the entire range of measured diameters.

Skyrmion motion is known to be affected by the grain-to-grain variation in the magnetic properties of polycrystalline metallic films[28]. Modelling predicts that the skyrmion Hall angle approaches the pristine film value as the velocity increases[27,30]. A suppression of both the skyrmion Hall angle and velocity is expected when the skyrmions sizes is comparable to the magnetic grain size[30]. In that micromagnetic model, both the angle and the velocity depend on the ratio of the grain size to the skyrmion diameter. Therefore, our results are not consistent with the established dynamical picture, as pinning from small grains would reduce both velocity and $\theta_{Sky}$ in a correlated way.

Figure 4 shows the correlation of $\theta_{Sky}$ with the skyrmion velocity. The raw velocity and corresponding $\theta_{Sky}$ data were binned into 1 ms$^{-1}$ and 5° intervals, respectively (see Fig. 4a). The average $\theta_{Sky}$ and the number of skyrmions for each bin is shown in Fig. 4b, c, respectively. Figure 4d–l show the individual data sets acquired for different applied field strengths. The overall average velocity is $6 \pm 1$ ms$^{-1}$. The majority of the observed skyrmions were observed to move at similar velocities close to this average value (see the histogram in Fig. 3c). Variations in the velocity were mostly observed at diameters below 50 nm and above 750 nm. However, it is worth noting that only ~9% out of all the moving skyrmions move with velocities outside the 5 to 7 ms$^{-1}$ range (i.e., 59 out of a total of 680 skyrmions). A large spread in the skyrmion Hall angle is observed at velocities below 3 ms$^{-1}$, with angles ranging from −80° to +80° (see Fig. 4a). This spread is much less at higher velocities. Nevertheless, the average value of $\theta_{Sky}$ is unaffected by the velocity. A non-zero skyrmion Hall angle is a theoretically predicted behaviour for skyrmion motion in a disordered system in the flow regime . In this regime, both moving and pinned skyrmions coexist, and large fluctuations of the velocity transverse to the driving force are expected resulting in a large scatter in the observed skyrmion Hall angle[23]. Increasing the velocity reduces the scatter of the motion and hence the scatter of the skyrmion Hall angle (as consistent with data in Fig. 4a).

**Oersted field effects**. The flow of a current through a wire generates an Oersted field. This leads to an additional out-of-plane field which varies in magnitude and sign across the $x$ direction of the wire. Following the Biot-Savart law, and integrating over the dimensions of the nanowire, the magnetic field in the wire can be computed by,

$$H_z(x', z') = \frac{I}{8\pi w t} \int_{-w/2}^{w/2} \int_{-t/2}^{t/2} \left( \frac{z' - z}{(x - x')^2 + (z - z')^2} \right) dx\, dz, \quad (3)$$

where $w$ is the wire width, and $t$ is the wire thickness of the device. Figure 5a shows the spatial variation across the wire of the

$z$ component of the Oersted field evaluated at a height of 1 nm above the centre of the wire. At the edges of the wire a maximum field of $\pm 13$ mT is reached. The measured coercive field at room temperature of the unpatterned material system is 23 mT (see Fig. 1a). During the current pulse the wire heats up and the coercive field is expected to decrease, hence it is conceivable that the Oersted field affects the magnetic domains.

Figure 5b shows the average diameter of the skyrmions throughout 50 nm vertical slices taken along the wire (see inset in Fig. 5a) at each applied field. Experimentally, the effect of this Oersted field can be seen in the slight decrease of the diameter as the skyrmions move across the wire. The skyrmion Hall angle leads to a transverse component to the motion, and hence a moving skyrmion will traverse through a change in the $z$ component of the field, i.e., skyrmions on the right-hand side experience a more negative applied field. Fitting straight lines to the data in Fig. 5b results in the confirmation of a negative slope (see Fig. 5c) when a static field in the range of −0.5 to −3 mT is applied. At zero field, the magnetic structures in the wire remains unaffected. At fields more negative than −3 mT, the spatial distribution of the skyrmion diameter indicates skyrmion growth with increasing field. This anomaly is consistent with the anomaly in the average diameter data, and most likely also an artefact of the skyrmion annihilation events. Skyrmion annihilation occurs frequently, however, annihilation events are not counted as finite diameters and thus result in an artificial skewing of the data towards larger diameters (see Fig. 2b).

While the field gradient is observed to influencing the spatial distribution of the skyrmion diameter, which would result in a spatially varying skyrmion Hall angle as the skyrmion traverses the wire, it does not explain the observed diameter-independence. Recent experiments on B20 skyrmion crystals have shown that Bloch skyrmions can be controlled by non-uniform magnetic fields[34]. In the case of Néel skyrmions, field-gradient manipulation is theoretically predicted and should be diameter-dependent[35].

To quantify the effect of a magnetic field gradient on the skyrmion Hall angle, an additional force in the Thiele equation, $F_B$, was included. This force is perpendicular to the force $F_{stt}$ due to spin-transfer torque in the wire, leading to a corrected value of the skyrmion Hall angle:

$$\theta_H = \tan^{-1}\left[ \left( 1 + \frac{\alpha_G \mathfrak{D}}{Q} \frac{F_B}{F_{stt}} \right) \Big/ \left( \frac{F_B}{F_{stt}} - \frac{\alpha_G \mathfrak{D}}{Q} \right) \right], \quad (4)$$

which now depends on the ratio $F_B/F_{stt}$. In the limit $F_B \ll F_{stt}$, Eq. 4, $\tan\theta_{Sky} = -Q/\alpha_G\mathfrak{D}$, is recovered. As the Thiele equation is first-order in time, the ratio $F_B/F_{stt}$ is $v_B/v_{stt}$. In order to estimate an upper bound for this parameter, the speed at which a field gradient of magnitude 5 mT μm$^{-1}$ drives skyrmions was investigated by micromagnetic modelling using the Fidimag package[36]. The time that a skyrmion took to travel 500 nm in a field gradient of this magnitude was measured using a discretisation of 2.5 nm × 2.5 nm × 2.5 nm. All material parameters were set to the values previously derived in this paper except for the uniaxial anisotropy, which was reduced to 0.1 MJm$^{-3}$ to ensure the stability of Néel skyrmions in the magnetic field regions of interest in the absence of the demagnetising field. Out-of-plane net magnetic fields in the range of 3–5 mT were used to derive a mean value of $F_B/F_{stt} = 0.0028 \pm 0.0001$. The small value of this parameter indicates that the force due to a magnetic field gradient is negligible in this case, and that the Hall angle should be well approximated by Eq. 2.

**Skyrmion energy landscape**. The data shown in Fig. 2c–l suggests that the skyrmions tend to follow distinct pathways through

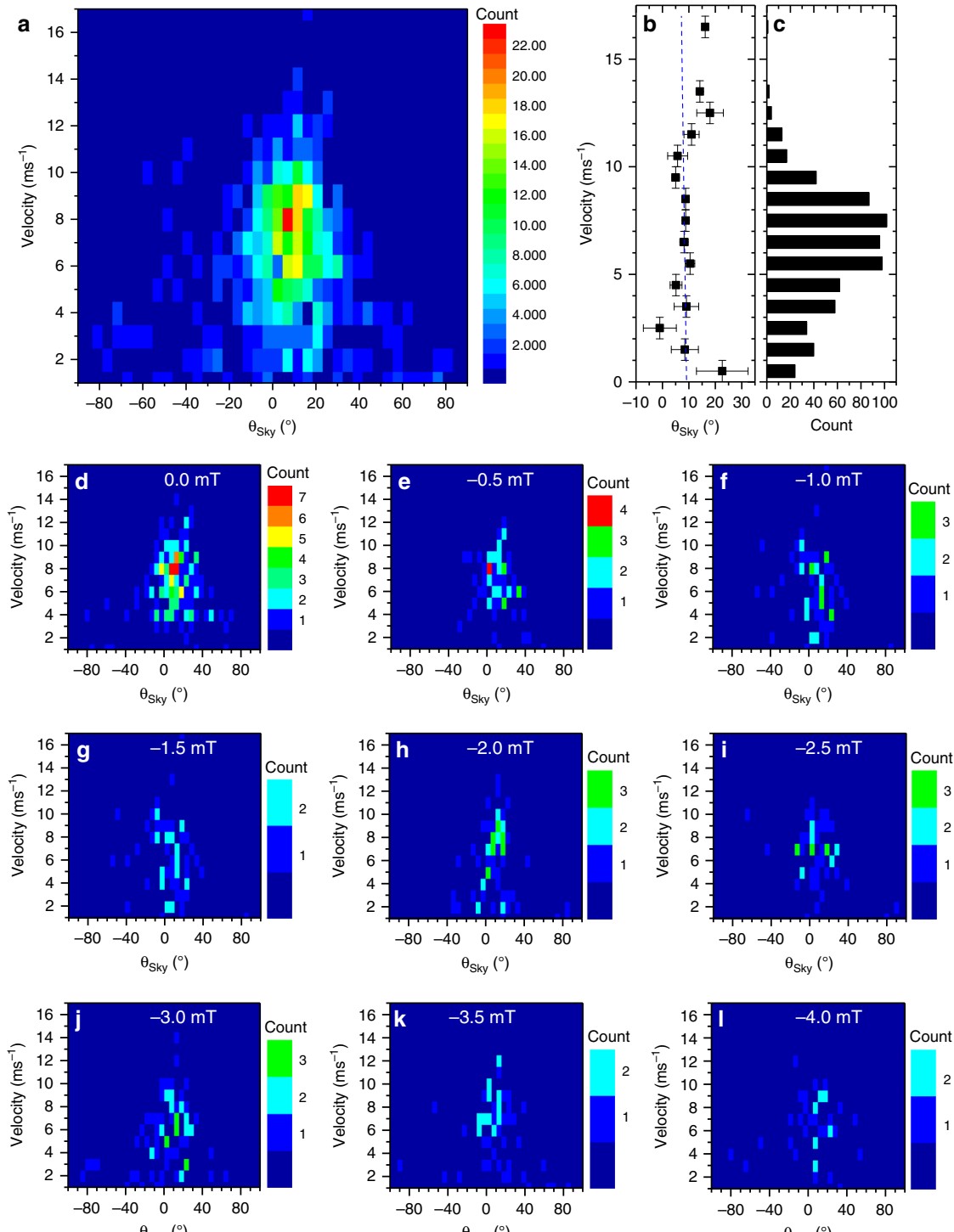

**Fig. 4 Relationship between skyrmion Hall angle and skyrmion velocity. a** Frequency count of skyrmion velocity versus skyrmion Hall angle $\theta_{Sky}$. The velocity was calculated from the distance travelled, as taken from the skyrmion core position, between consecutive images, i.e., in 18 ns. The velocity and angles were binned into $1\,ms^{-1}$ and $5°$ intervals, respectively. **b** Average $\theta_{Sky}$ versus velocity evaluated from data in **a** with the respective histogram **c**. Blue dashed line shows linear fit to data. Standard errors are plotted. **d–l** shows velocity-skyrmion Hall angle plots for different applied field. Taken together, these plots will give the overall plot in **a**.

the wire irrespective of their size. For instance, the skyrmion tracked by the navy hexagons and the skyrmion tracked by the purple triangles follow the same path in Fig. 2f. Furthermore, distinct track changes were observed, for example in Fig. 2e, where the skyrmion tracked by the navy hexagons and that tracked by the brown downwards triangles, change their

trajectory. An intensity map made by averaging all STXM images shows a very uneven distribution of skyrmion positions throughout the wire (see Fig. 6a). The intensity of each image was normalised to the same non-magnetic region and the images then were averaged together. Bright areas represent areas with a high probability of being occupied by a skyrmion, either pinned or

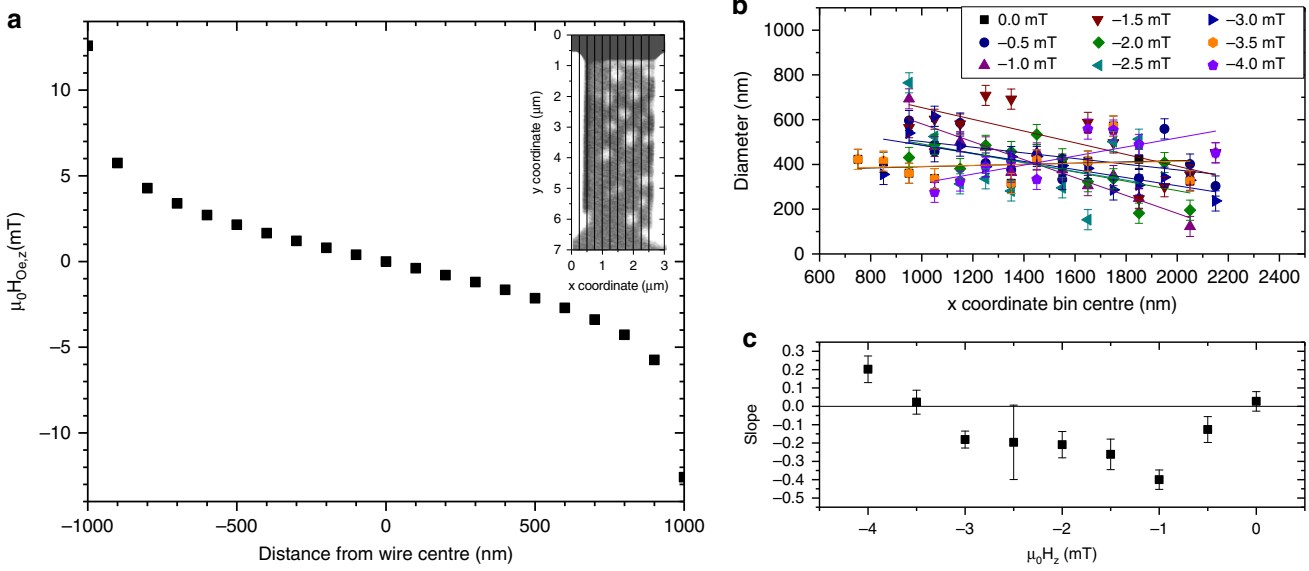

**Fig. 5 Oersted field across the wire generated by the current pulse. a** Oersted field, $\mu_0 H_{Oe,z}$ calculated for a current in a rectangular wire with a current density of $5.6 \times 10^{11}$ Am$^{-2}$, at a level 1 nm above the wire centre. **b** Average skyrmion diameter distributed across 50 nm segments (as illustrated in Fig. 5a inset) with respect to the applied field (Lines show the best linear fit). Error bars reflect the image resolution. **c** Slope versus field of a line fitted to the data shown in **b**. The line plotted is a guide to the eye. A negative slope reflects skyrmions shrinking in diameter as they traverse the wire from left to right. This is a consequence of the Oersted field generated by the current flowing through the wire. Error bars extracted from fit.

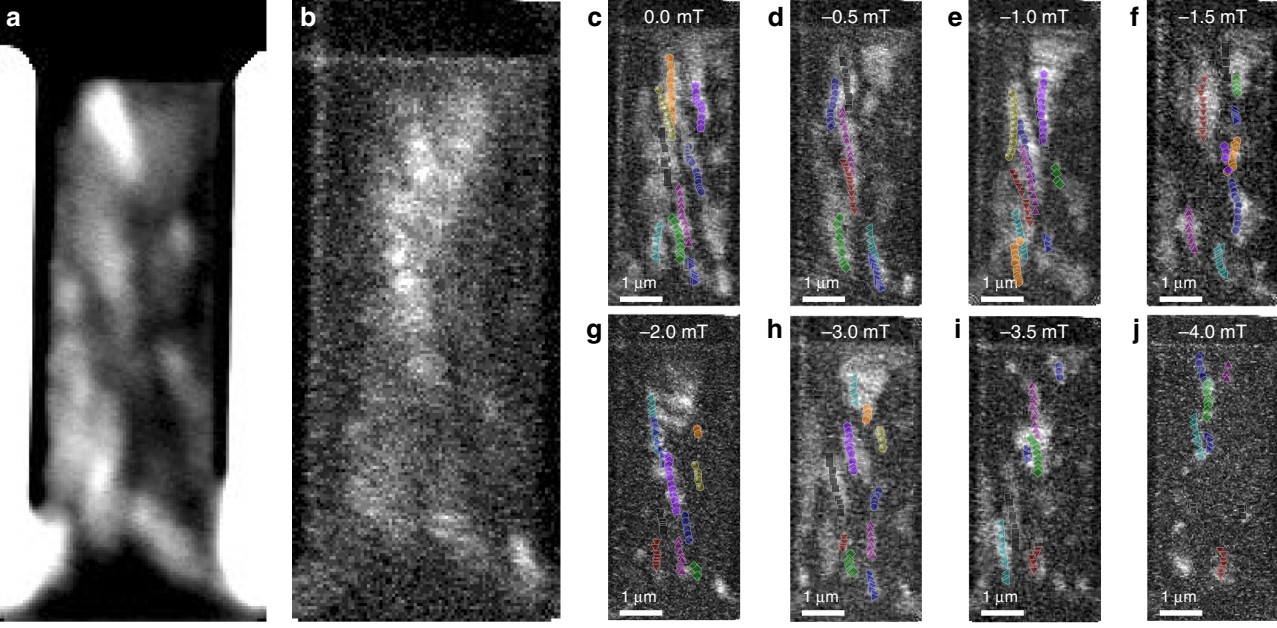

**Fig. 6 Skyrmion motion maps. a** Average intensity map of all STXM images. Bright regions represent areas with high probability of containing a reversed magnetic domain or skyrmion. **b** Average intensity map of the absolute difference between consecutive images. Bright regions show high probability of an expanding magnetic domain or a moving skyrmion. **c–j** Average intensity map of the absolute difference between consecutive images obtained at each applied field, superimposed are the skyrmion centre positions (coloured symbols) throughout the pulse series from Fig. 2.

movable. Figure 6b is an average intensity map of the differences between two consecutive images. Bright areas indicate regions where a magnetic change has occurred due to the current pulse, either skyrmion growth or motion. Figure 6c–j show the skyrmion centre positions throughout the excitation sequence at each applied field superimposed onto the average intensity images obtained at each applied field. A change in the skyrmion centre represents a motion event and thus by combining the average intensity map with the skyrmion centre motion one can distinguish growth from motion events.

Not all nucleated skyrmions were observed to move. Out of 197 nucleated skyrmions only 85 (44%) were seen to move in response to current pulses. Figure 7a shows skyrmion nucleation spots as open dark blue circles. The overlaid solid light blue circles highlight those skyrmions which then moved.

Figure 7b–d shows examples of phenomena that were observed to affect the skyrmion trajectories. We identify two primary reasons for changes in the skyrmion trajectory: Deflection from pinning sites and deflection from neighbouring magnetic domains or skyrmions. Skyrmions (1), (3) and (4) in Fig. 7b–d

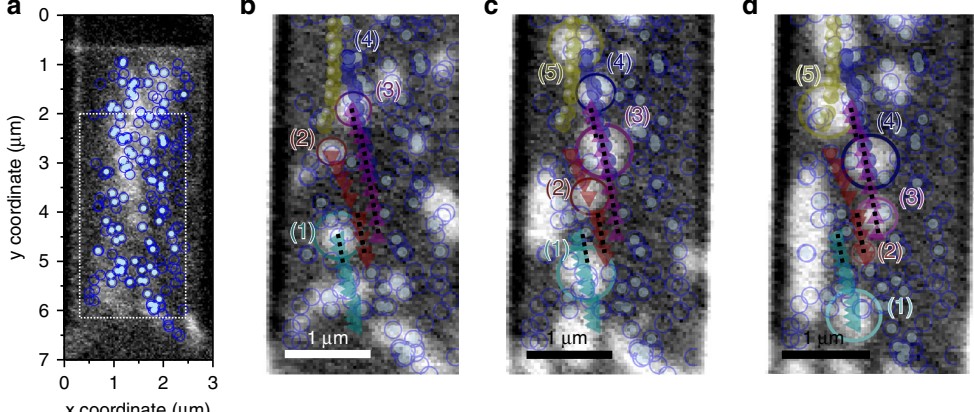

**Fig. 7 Energy landscape dominated motion. a** Skyrmion nucleation sites superimposed on the map shown in b (open blue circles) versus nucleation site of moving skyrmion (full light blue circles). **b–d** Zoom in of the area surrounded by the white dotted square in **a**. The centre positions of selected skyrmions imaged at −1 mT are superimposed as solid, spheres, triangles and hexagons, respectively. Changes to the centre trajectory as the skyrmions travel through the wire are apparent. Trajectory changes were observed to occur due to pinning sites (1), (3) and (4) and skyrmion–skyrmion (5) or skyrmion-magnetic domain deflection (2). When local effects do not dominate, the skyrmions follow a trajectory around 10° (dotted black line) which is consistent with the angles predicted by the Thiele equation for skyrmions with a diameter larger than 250 nm.

change trajectory abruptly due to pinning sites. It is interesting to note that the majority of the 18 skyrmions with a diameter smaller than 150 nm were observed in the initial, nucleation image and hence by default are likely to be in regions with defects. Skyrmions (2) and (5) change trajectory smoothly due to deflection from other magnetic domains and skyrmions. In areas of low pinning, i.e., regions with a low skyrmion nucleation probability[37], skyrmions are seen to move with an angle around 10°. This angle is consistent with predictions using the modified Thiele equation, for skyrmions with diameters above 250 nm (as predicted by Eq. 1). Hence, this is a consequence of the skyrmion's non-trivial topology. This experimentally confirms that the local energy landscape of multilayer skyrmion systems not only changes the drive dependence of the skyrmion Hall angle[20–26] but also quenches its predicted diameter dependence.

We have shown that Pt/CoB/Ir multilayers support skyrmions at zero field and that they can be moved by an electrical current. A 9-ns-long current pulse with a peak current density of $5.6 \times 10^{11}$ Am$^{-2}$ was shown to displace skyrmions with an average velocity of $6 \pm 1$ ms$^{-1}$. While the multilayers studied here were not optimised for current to spin-current conversion and spin orbit torque efficiency, the observed skyrmion motion reversal under current direction inversion indicates that spin-orbit torques are operative. A diameter-independent skyrmion Hall angle of $9° \pm 2°$ was observed. While the skyrmion Hall angle is consistent with the predictions of the rigid skyrmion Thiele picture for skyrmions with diameters above 175 nm, the overall diameter independence is quite surprising. The large, non-zero, velocity-dependent scatter of the Hall angle is consistent with the flow regime in which moving and pinned skyrmions coexist. The skyrmion trajectories are influenced by pinning sites and by skyrmion–skyrmion interactions. In areas of low pinning and skyrmion density the expected skyrmion Hall angle is recovered. Tailoring of the local magnetic parameters, introducing low pinning pathways, can therefore be envisioned to transport skyrmions down parallel lanes within the nanowire device in a robust manner, providing a barrier to the innate transverse motion skyrmions exhibit when driven by spin torques. By tracking the motion of skyrmions and correlating it to nucleation sites local changes in the magnetic energy landscape are inferred–allowing for its engineering in future device applications.

## Methods

**Multilayer growth and wire fabrication**. The thin films were deposited using DC magnetron sputtering at a base pressure of $2 \times 10^{-8}$ mbar and a target-substrate separation of ~7 cm. During the growth the argon pressure was 3.2 mbar. Typical growth rates of around 0.1 nms$^{-1}$ were used. The superlattice stack, $[Co_{68}B_{32}$ (0.8)/ Ir (0.4)/Pt (0.6)]$_{\times 5}$, (thicknesses are in nm), was grown on a seed layer of 3.2 nm Ta/2.7 nm Pt and capped with 2.2 nm of Ta. The patterned structures were grown on 200-nm-thick, highly resistive silicon nitride membranes (Silson Ltd, Warwickshire, UK). An identical thin film was simultaneously sputtered onto a thermally oxidised Si substrate (with an oxide layer thickness of 100 nm) to allow for the characterisation of the materials properties using standard techniques. X-ray reflectivity was used to measure the layer thicknesses, and room temperature polar magneto-optical Kerr effect (MOKE) magnetometry and in-plane superconducting quantum interference device-vibrating sample magnetometry (SQUID-VSM) were used to confirm the out-of-plane easy axis of the superlattice and to extract the saturation magnetisation of the $Co_{68}B_{32}$ ferromagnetic layer ($M_S = 1.2 \pm 0.1$ M Am$^{-1}$).

The 2-μm-wide wires were fabricated using electron-beam lithography with a positive resist lift-off process. The resist layer consisted of a bilayer of electron-beam-sensitive, positive, resist: the bottom layer was methyl-methacrylate (MMA) and the top layer was polymethylmethacrylate (PMMA). The spun and baked resist bilayer was exposed using a 100 kV Vistec EBPG 5000Plus electron beam writer with a writing dose of 1650 μC cm$^{-2}$. Following the exposure, the devices were developed for 90 s in a 1:3 methyl-isobutyl-ketone and isopropyl alcohol solution (by volume) and rinsed for 60 s in isopropyl alcohol. After the heterostructure was deposited, the unpatterned regions were lifted off in acetone. Finally, thermally evaporated, 200-nm-thick Cu electrodes were fabricated again by lift off and were designed to achieve an electrical impedance close to 50 Ohm minimising unwanted reflections of the injected current pulses.

**Brillouin light scattering**. Brillouin light scattering was used to measure the Dzyaloshinskii–Moriya Interaction (DMI) strength, $D$, of the continuous heterostructure sample. $D$ is extracted by measuring the asymmetry in the Stokes and anti-Stokes lines of light that has been inelastically scattered from propagating spin waves. The DMI strength is directly proportional to the frequency shifts of the inelastically scattered light with respect to the incident laser beam frequency,

$$\Delta f = f_S - f_{AS} = \frac{2\gamma}{\pi M_S} Dk, \qquad (5)$$

where $k$ is the absolute value of the magnon wavevector, $f_S$ is the Stokes frequency, $f_{AS}$ is the anti-Stokes frequency, and $\gamma = 190$ GHzT$^{-1}$ is the gyromagnetic ratio. Spin waves of a given wavelength, which are propagating in opposite directions in a sample with strong DMI, have different energies. This behaviour is known as propagation nonreciprocity and occurs in the Damon-Eshbach geometry. The Stokes and anti-Stokes spectra measured for $k = 7 \times 10$ μm$^{-1}$ and $k = 11.09$ μm$^{-1}$ can be seen in Fig. 1b. $D$ was calculated using Eq. 5 and was found to be $-1.1 \pm 0.1$ mJm$^{-2}$.

**Soft X-ray imaging**. The out-of-plane magnetisation of the devices was imaged using scanning transmission X-ray microscopy (STXM) at the PolLux (X07DA) beamline at the Swiss Light Source. The device was aligned perpendicular to the incident X-rays, which were tuned to the Co $L_3$ absorption edge (~778 eV). Out-of-plane magnetic contrast was achieved using X-ray magnetic circular dichroism, with the differential absorption being strong enough that STXM images were acquired using only a single photon helicity. A Fresnel zone plate with an outermost zone of 25 nm was used to focus the X-rays on the sample. A spatial resolution on the order of 25–30 nm was achieved. The images were taken with a pixel size of 35 nm. The images were acquired at room temperature. The skyrmions were nucleated using a 9-ns-long current pulse at a current density of ~$5 \times 10^{12}$ Am$^{-2}$. The nucleated skyrmions were then moved using current pulses with a duration of ~9 ns and a current density of $5.6 \times 10^{11}$ Am$^{-2}$. An arbitrary waveform generator (AWG) was used to create the applied sine wave pulse of ½ period. A broadband RF amplifier was employed to amplify the signal generated by the AWG. The absence of a DC component in the transfer function of the amplifier is responsible for the over/undershoot of the injected current pulses that can be observed in Fig. 1c. The current injected across the wires was measured with a 50 Ω terminated real-time oscilloscope. A static out-of-plane magnetic field was applied in the range between 0 and −4 mT. A negative field opposes the skyrmion core direction and hence compresses the skyrmion diameter $d$.

## Data availability

The data associated with this paper are openly available from https://doi.org/10.5518/742.

## Code availability

The codes associated with the simulations presented in this paper are openly available from https://doi.org/10.5518/742.

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

## Acknowledgements

Support from EP/M000923/1 and the European Union (H2020 grant MAGicSky No. FET-Open-665095.103 and EMPIR project TOPS "Metrology for topological spin structures" (#17FUN08)), as well as from the Diamond Light Source and the Swiss Nanoscience Institute (grant P1502), is gratefully acknowledged. T.H. gratefully acknowledges support from EPSRC (EP/N032128/1). R.B. acknowledges support from Diamond Light Source and the EPSRC through a Doctoral Training Grant. Part of this work was carried out at the PolLux (X07DA) beamline of the Swiss Light Source. Brillouin light scattering study in submicron LEGION based type of magnonic topology was supported by Russian Science Foundation (No. 18-79-00198). S.A.N. acknowledges support by the Government of the Russian Federation (Agreement No. 074-02-2018-286) and Russian Science Foundation (No. 19-19-00607). The PolLux end station was financed by the German Minister für Bildung und Forschung (BMBF) through contracts 05KS4WE1/6 and 05KS7WE1. This project has received funding from the EU-H2020 research and innovation programme under grant agreement N 654360 and has benefitted from the access provided by the Paul Scherrer Institute in Villigen within the framework of the NFFA-Europe Transnational Access Activity. Use was made of facilities within the Bragg Centre for Materials Research at Leeds.

## Author contributions

K.Z. and S.F. conceived the experiment, with input from J.R., G.B. and C.H.M. K.Z. optimised and grew the Pt/Co$_{68}$B$_{32}$/Ir multilayer stacks. K.Z. and S.F. designed the sample. K.Z. performed the magnetometry measurements of the multilayer stacks. A.V.S. and S.A.N. performed and analysed the BLS measurement. S.F., M.C.R. and E.H.L. lithographically patterned the samples. K.Z., S.F., J.M., A.J.H., G.B. and J.R. performed the STXM experiments. K.Z. analysed the experimental data. C.B. performed the Oersted

field simulation. R.B., T.H. and G.L. calculated and simulated the field gradient driving force on the skyrmions. K.Z. wrote the paper. All authors reviewed and contributed to the paper.

## Competing interests

The authors declare no competing interests.
