## [Peer Review File · Nature Communications]

Reviewers' comments:

Reviewer #1 (Remarks to the Author):

The MS by Zeissler et al reports an experimental study of the skyrmion Hall angle in chiral magnetic multilayers. They use STXM to determine the location of individual skyrmions as they apply subsequent electric field pulses. In contrast to theoretical expectations they observe that the skyrmion Hall angle seems to be independent from the diameter of the skyrmions. In particular, the deviation seems to happen for smaller skyrmion diameters, i.e. below 200nm

This is an interesting and timely study addressing an important topic in the physics of skyrmions. The paper is well written and the logical flow is easy to follow. The authors did a thorough and state of the art precharacterization to determine, e.g. Keff and the DMI strength using MOKE and BLS, resp.

The STXM experiments are also state of the art appropriate for this research.

Given the broad interest in the skyrmion community, I support the publication in Nauter Communications.

However, I have a few comments, that I would like to see the authors to address before this paper can be accepted for publication.

My major concern is that the deviation from the predicted behavior is at smaller diameters, where I would expect that both the imaging as well as the defect landscape has larger impact. The authors discuss the skyrmion energy landscape, and the conclusions might be justified, but I would like to see a bit more critical assessment here. For example, the fact that not all skyrmions moved seems to be related to pinning at defects. How do the authors know, where to draw the line between not moving, or just slowly moving, i.e. slower than the imaging times.

Minor comments:

Abstract: I suggest to mention the range of diameters that were studied here

p2: The formatting of D , the dissipative force seems to be inconsistent throughout the third paragraph.

p3: Results. The traces of positive and negative pulses seems to not go back to zero after the pulse is applied. Does that affect the measurements?

Reviewer #2 (Remarks to the Author):

In Pt/Co/Ir multilayers with a broken interfacial symmetry, Katharina Zeissler et al. studied the current-driven skyrmion dynamics. Specifically, authors observed an intriguing skyrmion diameter-independent skyrmion Hall effect.

Magnetic skyrmions are topological spin textures that contain many exciting topological physics, in addition to its potential application as information carrier in spintronic devices. Along this direction, one of the most important aspects is the controllable motion of skyrmion dynamics, which is the key for spintronic devices. It is known that the dynamics of skyrmions in multilayers driven by currents exhibits a skyrmion Hall effect, namely moving towards the side of the devices as a result of topological Magnus force. While this gives rise to meaningful physics, it hampers the potential application due to the presence of skyrmion-edge interaction.

On this very exciting topic, this manuscript reports a comprehensive study of skyrmion dynamics, in which the diameter-independent skyrmion Hall effect is observed. The data, from my point of views, is at the highest quality, and explanation is sufficient (pending on minor inputs). These three facts together ensure its publication in journals such as Nature Com. Below are my minor comments.

1: Considering the influence of driving forces to the skyrmion dynamics. When driving is weak, the dynamics of skyrmions is in the creep motion regime where the skyrmion dynamics can be stochastic. By contrast, the motion will be in the plastic flow motion regime in the strong driving regime. Such a study is conducted but not specifically discussed in the manuscript. Authors should make sufficient revisions.

2. From Ref. 20 and references therein, it is known that skyrmion Hall angle is dependent on the dissipative tensor D , which is also dependent on the skyrmion diameter. This is expected since larger skyrmion contribute strongly to dissipation and hence smaller skyrmion Hall angle for the fixed driving force. Thus, one would expect naturally a diameter-dependent skyrmion Hall effect. While authors made efforts, I felt that the current explanation is not clear yet.

3. Since the multilayers are very thick, the contribution from spin-orbit torques is thus minimized. Did authors consider this? Additionally, it is known that the skyrmion spin profile could exhibit a complex 3D textures, which could complicate its dynamics upon applying spin orbit torques. Moreover, the counteractive SOTs of Pt layers from the top and bottom could also contribute to some extent.

Reviewer #3 (Remarks to the Author):

Report for

Diameter-independent skyrmion Hall angle in the plastic flow regime observed in chiral magnetic multilayers Katharina Zeissler^{1,2*}, Simone Finizio³, Craig Barton², Alexandra Huxtable¹, Jamie Massey¹, Jörg Raabe³, Alexandr V. Sadovnikov^{4,5}, Sergey A. Nikitov^{4,5}, Richard Brearton^{6,7}, Thorsten Hesjedal⁶, Gerrit van der Laan⁷, Mark C. Rosamond⁸, Edmund H. Linfield⁸, Gavin Burnell¹, Christopher H. Marrows

Skyrmions are particle like magnetic textures that were discovered in 2009 and since then have been intensely studied. A wealth of new materials have now been found that support skyrmions, including systems where skyrmions can be stable at room temperature. Because skyrmions can be sent into motion, they are also promising candidates for a number of applications. In terms of basic science, skyrmions also represent a new class of system with collective interactions in the presence of quenched disorder. One of the most striking aspects of their dynamics is the strong non-dissipative component or the Magnus force. One consequence of the Magnus force is that the skyrmions will move at an angle with respect to the drive known as the skyrmion Hall effect. In principle the skyrmion Hall angle is independent of drive; however, as initially predicted in simulations, when pinning is present the skyrmion Hall angle becomes drive dependent. This effect has been observed in various experiments and there is some controversy over whether this effect is due to the skyrmion changing shape or to skyrmion-pin interactions. In this work the authors study the effect of the skyrmion diameter on the skyrmion Hall angle and find that it is independent. Additionally, the authors map out how the potential energy landscape affects the skyrmion motion. The work is very extensive and the authors had to do a fair amount of analysis which is nicely presented. Skyrmion dynamics experiments are rather difficult since they typically require imaging of the skyrmion motion. This work helps to elucidate the role of disorder on skyrmion dynamics and the Hall effect, and is not only of interest to workers in skyrmions but also for the broader audience studying statistical physics and nonequilibrium systems. I have a few points for the authors to address.

(1) Did the authors only drive the skyrmions in one direction or did they also reverse the drive? One could check whether the skyrmions followed the same trajectories on the reversed drive or if they had the same Hall angle. This could also give insight into the potential energy landscape. One

work that explored this was

"Reversible to irreversible transitions in periodically driven skyrmion systems",
B.L. Brown, C. Reichhardt and C.J.O. Reichhardt
New J. Phys. 21 013001 (2019).

(2) Are there any creep effects occurring in this work, or are thermal motions negligible?

(3) Just some points regarding the background of the system. The first predictions for a drive dependent skyrmion Hall angle were presented in 2015, where it was argued to be due to a pinning effect:

Collective Transport Properties of Driven Skyrmions with Random Disorder
C. Reichhardt, D. Ray, and C. J. Olson Reichhardt
Phys. Rev. Lett. 114, 217202 (2015)

Quantized transport for a skyrmion moving on a two-dimensional periodic substrate
C. Reichhardt, D. Ray, and C. J. Olson Reichhardt
Phys. Rev. B 91, 104426 (2015) - Published 26 March 2015

This was followed up by additional work

Fluctuations and noise signatures of driven magnetic skyrmions
Sebastián A. Díaz, C. J. O. Reichhardt, Daniel P. Arovas, Avadh Saxena, and C. Reichhardt
Phys. Rev. B 96, 085106 (2017)

C. Reichhardt and C. J. O. Reichhardt, Noise fluctuations and drive dependence of the skyrmion Hall effect in disordered systems, New J. Phys. 18, 095005 (2016).

"Thermal creep and the skyrmion Hall angle in driven skyrmion crystals",
C. Reichhardt and C.J.O. Reichhardt
J. Phys.: Condens. Matter 31 07LT01 (2019)

The paper cited in Ref

Lin, S., Reichhardt, C., Batista, C. D. & Saxena, A. Particle model for skyrmions in metallic chiral magnets : Dynamics , pinning , and creep. Phys. Rev. B87, 214419 (2013)

talks about pinning but not the skyrmion Hall effect.

Also regarding the "controversy", the only work that stated that it was the skyrmion shape or current effect that caused the skyrmion Hall angle to change was:

K. Litzius, I. Lemesh, B. Krüger, P. Bassirian, L. Caretta, K. Richter, F. Büttner, K. Sato, O. A. Tretiakov, J. Förster, R. M. Reeve, M. Weigand, I. Bykova, H. Stoll, G. Schütz, G. S. D. Beach, and M. Kläui, Skyrmion Hall effect revealed by direct time-resolved X-ray microscopy, Nat. Phys. 13, 170 (2017).

However, this work has no theory or calculation to support this claim; it was simply stated. Also there are now several continuum based simulations that see no change in the skyrmion Hall angle when pinning is absent.

"Diameter-independent skyrmion Hall angle in the plastic flow regime observed in chiral magnetic multilayers"

We are pleased to resubmit a revised version of our manuscript that is now entitled "Diameter-independent skyrmion Hall angle observed in chiral magnetic multilayers".

Please find below a point-by-point response to all three referee reports, interleaved in blue.

Reviewer #1 (Remarks to the Author):

The MS by Zeissler et al reports an experimental study of the skyrmion Hall angle in chiral magnetic multilayers. They use STXM to determine the location of individual skyrmions as they apply subsequent electric field pulses. In contrast to theoretical expectations they observe that the skyrmion HALL angle seems to be independent from the diameter of the skyrmions. In particular, the deviation seems to happen for smaller skyrmion diameters, i.e. below 200nm This is an interesting and timely study addressing an important topic in the physics of skyrmions. The paper is well written and the logical flow is easy to follow. The authors did a thorough and state of the art precharacterization to determine, e.g. Keff and the DMI strength using MOKE and BLS, resp. The STXM experiments are also state of the art appropriate for this research.

Given the broad interest in the skyrmion community, I support the publication in Nauter Communications.

However, I have a few comments, that I would like to see the authors to address before this paper can be accepted for publication.

We thank the Reviewer for his/her positive comments, which we address below.

My major concern is that the deviation from the predicted behavior is at smaller diameters, where I would expect that both the imaging as well as the defect landscape has larger impact. The authors discuss the skyrmion energy landscape, and the conclusions might be justified, but I would like to see a bit more critical assessment here. For example, the fact that not all skyrmions moved seems to be related to pinning at defects. How do the authors know, where to draw the line between not moving, or just slowly moving, i.e. slower than the imaging times.

The referee is correct that the departures from the predictions of the rigid-skyrmion theory appear for small diameters (our figure 3b). While skyrmions stabilised in these CoB multilayers were observed to predominantly have diameters larger than 180 nm, 30 events were observed to deviate from the predicted behaviour with $\alpha=0.07$. For the highly damped limit, over 650 events do not align with the ideal case predictions. However, the observed data is best described by a straight line with a slope of zero, which differs from the predictions of this theory for any value of damping considered here. This is because the rigid-skyrmion theory is based on the Thiele approach and does not include pinning forces arising from the energy landscape, and that the discrepancy between theory and experiment is due to these forces is the gist of our manuscript.

We do not believe that there are imaging artefacts at any diameter. The time to acquire a single STXM image with XMCD was of the order of 5 minutes. No current pulses were applied during the image acquisition and the applied fields were chosen to be well below the coercive field of the

microstructure, and so the state of the sample is not changing during its imaging. This was checked experimentally: successive imaging without the application of current pulses showed no change in the magnetic domains. Therefore, the skyrmion motion is tracked by looking at the static state before and after the current pulse event, and hence the chosen imaging time has no implications for the motion detected. The motion is determined by a change in the skyrmion centre position as determined by the Trackmate algorithm. Please see fig R1 for a typical example of the tracking result of the algorithm.

The manuscript was clarified in the

following way:

“An XMCD-STXM image was taken after two consecutive current pulses separated by a delay of 2 μ s. The centre of the displaced skyrmions was identified and tracked using the ImageJ TrackMate algorithm³⁴.”

was changed to

“A static XMCD-STXM image was taken after two consecutive current pulses separated by a delay of 2 μ s. The centre of the displaced skyrmions was identified and tracked using the ImageJ TrackMate algorithm³⁴. The skyrmion displacement and velocity was calculated from the change in the skyrmion centre coordinates identified by the algorithm before and after each current pulse pair.”

Minor comments:

Abstract: I suggest to mention the range of diameters that were studied here

Thank you very much for the suggestions. We changed the following sentence in the abstract

“In contrast, our experimental study finds that within the plastic flow regime the skyrmion Hall angle is diameter-independent.”

to

“In contrast, our experimental study finds that the skyrmion Hall angle is diameter-independent for skyrmions with diameters ranging from 35 to 825 nm”

p2: The formatting of D , the dissipative force seems to be inconsistent throughout the third paragraph.

We thank the referee for spotting this inconsistency. We have rectified this in the revised manuscript.

p3: Results. The traces of positive and negative pulses seems to not go back to zero after the pulse is applied. Does that affect the measurements?

All the skyrmion motion analysis on which our quantitative conclusions are based was performed on the data shown in figure 2. In that case a more optimised current pulse (see inset in figure 2 b) was applied. There is no overshoot observable here, and hence there is no detrimental effect on the measurements and analysis shown in the subsequent figures. Please note that in figure 1 there is indeed an observable overshoot with an amplitude of around 5×10^{11} A/m² and a duration of tens of ns. It is conceivable that this could indeed affect the measurement, but these data are not after that point considered in the manuscript. However, as a directional dependence is observed when the current direction is reversed, we conclude that the overshoot must be just below the threshold for skyrmion motion.

Reviewer #2 (Remarks to the Author):

In Pt/Co/Ir multilayers with a broken interfacial symmetry, Katharina Zeissler et al. studied the current-driven skyrmion dynamics. Specifically, authors observed an intriguing skyrmion diameter-independent skyrmion Hall effect.

Magnetic skyrmions are topological spin textures that contain many exciting topological physics, in addition to its potential application as information carrier in spintronic devices. Along this direction, one of the most important aspects is the controllable motion of skyrmion dynamics, which is the key for spintronic devices. It is known that the dynamics of skyrmions in multilayers driven by currents exhibits a skyrmion Hall effect, namely moving towards the side of the devices as a result of topological Magnus force. While this gives rise to meaningful physics, it hampers the potential application due to the presence of skyrmion-edge interaction.

On this very exciting topic, this manuscript reports a comprehensive study of skyrmion dynamics, in which the diameter-independent skyrmion Hall effect is observed. The data, from my point of views, is at the highest quality, and explanation is sufficient (pending on minor inputs). These three facts together ensure its publication in journals such as Nature Com. Below are my minor comments.

We thank the Reviewer for the positive review and the support for our manuscript. Below we have address the minor comments raised by the Reviewer.

1: Considering the influence of driving forces to the skyrmion dynamics. When driving is weak, the dynamics of skyrmions is in the creep motion regime where the skyrmion dynamics can be stochastic. By contrast, the motion will be in the plastic flow motion regime in the strong driving regime. Such a study is conducted but not specifically discussed in the manuscript. Authors should make sufficient revisions.

The plastic flow regime is defined by a linear response of the ratio of the perpendicular and parallel skyrmion velocity components to changes in the driving force. This does not necessarily mean there is a linear relationship between the skyrmion Hall angle and driving force [C. Reichhardt, et al., J. Phys.: Condens. Matter **31**, 07LT01 (2019)]. In the creep regime the perpendicular component of the skyrmion Hall angle is zero and hence a zero skyrmion Hall angle is observed i.e. skyrmions move parallel to the driving force. The limited synchrotron time and the small dynamical range between skyrmion motion and catastrophic damage to the multilayers due to heating effects unfortunately

means that not enough data was acquired to confirm that we indeed operated in the flow regime. What we know for certain is that no creep of the skyrmions is observed without the application of a current pulse, which indicates that the pinning potentials are much higher than the thermal energy of the system. However, it is important to note that the temperature within the multilayer during the current pulse is elevated and can reach temperatures close to the Curie temperature [S. Finizio, et al., Nano Letters **19**, 10 (2019)] and hence the transition between the creep regime and the elastic and plastic flow regime is not necessarily clear cut. The large scatter of the non-zero Hall angle is consistent with the flow regime [C. Reichhardt et al. New J. Phys. **18**, 095005 (2016)]. We also note that the skyrmion Hall angle is predicted to go to zero in the creep regime [C. Reichhardt, et al., J. Phys.: Condens. Matter **31**, 07LT01 (2019)]. By observing a constant skyrmion Hall angle of 9° we are most likely operating outside the creep regime. Furthermore, we are operating at relative high speeds of 6 m/s in comparison with the speed usually associated with the creep regime. As the driving force throughout the experiment was kept constant the conclusions drawn by the presented data are valid irrespective of the position in the motion phase diagram. We have therefore removed all references to the plastic flow regime and changed the following two statements from:

“Nevertheless, a large spread in the skyrmion Hall angle is observed at velocities below 3 m/s, with angles ranging from -80° to $+80^\circ$ (see figure 4 a). This spread is much less at higher velocities. This is a theoretically predicted behaviour for skyrmion motion in a disordered system in the plastic flow regime²⁶. In this regime, both moving and pinned skyrmions coexist, and large fluctuations of the velocity transverse to the driving force are expected resulting in a large scatter in the observed skyrmion Hall angle.”

To

“A large spread in the skyrmion Hall angle is observed at velocities below 3 m/s, with angles ranging from -80° to $+80^\circ$ (see Fig. 4 a). This spread is much less at higher velocities. Nevertheless, the average value of θ_{sky} is unaffected by the velocity. A non-zero skyrmion Hall angle is a theoretically predicted behaviour for skyrmion motion in a disordered system in the flow regime. In this regime, both moving and pinned skyrmions coexist, and large fluctuations of the velocity transverse to the driving force are expected resulting in a large scatter in the observed skyrmion Hall angle²³.”

and

“The large, velocity-dependent scatter of the Hall angle is consistent with the plastic flow regime in which moving and pinned skyrmions coexist.”

to

“The large, non-zero, velocity-dependent scatter of the Hall angle is consistent with the flow regime in which moving and pinned skyrmions coexist.”

2. From Ref. 20 and references therein, it is known that skyrmion Hall angle is dependent on the dissipative tensor D , which is also dependent on the skyrmion diameter. This is expected since larger skyrmion contribute strongly to dissipation and hence smaller skyrmion Hall angle for the fixed driving force. Thus, one would expect naturally a diameter-dependent skyrmion Hall effect. While authors made efforts, I felt that the current explanation is not clear yet.

Thank you very much for this comment. The referee is right that, in the ideal case, i.e., if there are no imperfections, grains and other growth defects such as small spatial variations in the material thicknesses, the skyrmion Hall effect is predicted to be diameter dependent by the Thiele formulation, which treats the skyrmion as a rigid particle-like object subject to gyrotropic,

dissipative, and driving forces. However, imperfections and the resulting interaction of skyrmions with pinning sites have been shown experimentally to change the skyrmions response to external driving forces [W. Jiang, et al., Nat. Phys. **13**, 162 (2017), K. Litzius, et al., Nat. Phys. **13**, 170 (2017), R. Juge, et al. Phys Rev Appl. **12**, 044007 (2009)], which can be interpreted as the introduction of pinning forces arising from the energy landscape into a theory in which they were previously absent. Here we show that they furthermore suppress the expected diameter dependence of the skyrmion Hall angle. We have emphasised this point in the manuscript by changing the following:

“This experimentally confirms that in the low-velocity, plastic flow regime the skyrmion Hall angle is dominated by the local energy landscape of the nanowire and as such the theoretically predicted diameter dependence is quenched”

to

“This experimentally confirms that the local energy landscape of multilayer skyrmion systems not only changes the drive dependence of the skyrmion Hall angle²⁰⁻²⁷ but also quenches its predicted diameter dependence.”

3. Since the multilayers are very thick, the contribution from spin-orbit torques is thus minimized. Did authors consider this? Additionally, it is known that the skyrmion spin profile could exhibit a complex 3D textures, which could complicate its dynamics upon applying spin orbit torques. Moreover, the counteractive SOTs of Pt layers from the top and bottom could also contribute to some extent.

We thank the referee for this remark, and, we indeed considered this. While the multilayers studied here were not optimised for SOT efficiency and current to spin-current conversion (but rather for the ability to support skyrmions at zero field), we nevertheless observe motion in the order of 6 m/s which reverses under the reversal of the current direction, indicating that spin orbit torques are operative. With regards to complex 3D textures which can be observed in multilayer structures of >10 repetitions [W. Legrand, et al., Science Advances **4**, 7, (2018)] we would like to draw the attention of the referee to the ×5 repeat structure measured here. In films with such a low number of repetitions we do not expect these effects to occur.

We added the following sentence to the conclusion:

“While the multilayers studied here were not optimised for current to spin-current conversion and spin-orbit torque efficiency, the observed skyrmion motion reversal under current direction inversion indicates that spin-orbit torques are operative.”

Reviewer #3 (Remarks to the Author):

Report for

Diameter-independent skyrmion Hall angle in the plastic flow regime observed in chiral magnetic multilayers Katharina Zeissler^{1,2*}, Simone Finizio³, Craig Barton², Alexandra Huxtable¹, Jamie Massey¹, Jörg Raabe³, Alexandr V. Sadovnikov^{4,5}, Sergey A. Nikitov^{4,5}, Richard Brearton^{6,7}, Thorsten Hesjedal⁶, Gerrit van der Laan⁷, Mark C. Rosamond⁸, Edmund H. Linfield⁸, Gavin Burnell¹, Christopher H. Marrows

Skyrmions are particle like magnetic textures that were discovered in 2009 and since then have been intensely studied. A wealth of new materials have now been found that support skyrmions, including systems where skyrmions can be stable at room temperature. Because skyrmions can be sent into motion, they are also promising candidates for a number of applications. In terms of basic science, skyrmions also represent a new class of system with collective interactions in the presence of quenched disorder. One of the most striking aspects of their dynamics is the strong non-dissipative component or the Magnus force. One consequence of the Magnus force is that the skyrmions will move at an angle with respect to the drive known as the skyrmion Hall effect. In principle the skyrmion Hall angle is independent of drive; however, as initially predicted in simulations, when pinning is present the skyrmion Hall angle becomes drive dependent. This effect has been observed in various experiments and there is some controversy over whether this effect is due to the skyrmion changing shape or to skyrmion-pin interactions. In this work the authors study the effect of the skyrmion diameter on the skyrmion Hall angle and find that it is independent. Additionally, the authors map out how the potential energy landscape affects the skyrmion motion. The work is very extensive and the authors had to do a fair amount of analysis which is nicely presented. Skyrmion dynamics experiments are rather difficult since they typically require imaging of the skyrmion motion. This work helps to elucidate the role of disorder on skyrmion dynamics and the Hall effect, and is not only of interest to workers in skyrmions but also for the broader audience studying statistical physics and nonequilibrium systems. I have a few points for the authors to address.

We thank the Reviewer for his/her kind words, and the very useful comments, which we address below.

(1) Did the authors only drive the skyrmions in one direction or did they also reverse the drive? One could check whether the skyrmions followed the same trajectories on the reversed drive or if they had the same Hall angle. This could also give insight into the potential energy landscape. One work that explored this was

**"Reversible to irreversible transitions in periodically driven skyrmion systems"
B.L. Brown, C. Reichhardt and C.J.O. Reichhardt
New J. Phys. 21 013001 (2019).**

We reversed the driving force, however, only in the context of verifying that it is SOT that is driving the skyrmion motion. We agree this is a very interesting avenue for future work, however, as this involves extensive synchrotron time to collect such a full data set, is beyond the scope of this paper.

(2) Are there any creep effects occurring in this work, or are thermal motions negligible?

The plastic flow regime is defined by a linear response of the ratio of the perpendicular and parallel skyrmion velocity components to changes in the driving force. This does not necessarily mean there is a linear relationship between the skyrmion Hall angle and driving force [C. Reichhardt, et al., J. Phys.: Condens. Matter **31**, 07LT01 (2019)]. In the Creep regime the perpendicular component of the skyrmion Hall angle is zero and hence a zero skyrmion Hall angle is observed i.e. skyrmions move parallel to the driving force. The limited synchrotron time and the small dynamical range between skyrmion motion and catastrophic damage to the multilayers due to heating effects unfortunately means that not enough data was acquired to confirm that we indeed operated in the flow regime. What we know for certain is that no creep of the skyrmions is observed without the application of a current pulse, which indicates that the pinning potentials are much higher than the thermal energy

of the system. However, it is important to note that the temperature within the multilayer during the current pulse is elevated and can reach temperatures close to the Curie temperature [S. Finizio, et al., Nano Letters **19**, 10 (2019)] and hence the transition between the creep regime and the elastic and plastic flow regime is not necessarily clear cut. The large scatter of the non-zero Hall angle is consistent with the flow regime [C. Reichhardt et al. New J. Phys. **18**, 095005 (2016)]. We also note that the skyrmion Hall angle is predicted to go to zero in the creep regime [C. Reichhardt, et al., J. Phys.: Condens. Matter **31**, 07LT01 (2019)]. By observing a constant skyrmion Hall angle of 9° we are most likely operating outside the creep regime. Furthermore, we are operating at relative high speeds of 6 m/s in comparison with the speed usually associated with the creep regime. As the driving force throughout the experiment was kept constant the conclusions drawn by the presented data are valid irrespective of the position in the motion phase diagram. We have therefore removed all references to the plastic flow regime and changed the following two statements from:

“Nevertheless, a large spread in the skyrmion Hall angle is observed at velocities below 3 m/s, with angles ranging from -80° to $+80^\circ$ (see figure 4 a). This spread is much less at higher velocities. This is a theoretically predicted behaviour for skyrmion motion in a disordered system in the plastic flow regime²⁶. In this regime, both moving and pinned skyrmions coexist, and large fluctuations of the velocity transverse to the driving force are expected resulting in a large scatter in the observed skyrmion Hall angle.”

To

“A large spread in the skyrmion Hall angle is observed at velocities below 3 m/s, with angles ranging from -80° to $+80^\circ$ (see Fig. 4 a). This spread is much less at higher velocities. Nevertheless, the average value of θ_{sky} is unaffected by the velocity. A non-zero skyrmion Hall angle is a theoretically predicted behaviour for skyrmion motion in a disordered system in the flow regime. In this regime, both moving and pinned skyrmions coexist, and large fluctuations of the velocity transverse to the driving force are expected resulting in a large scatter in the observed skyrmion Hall angle²³.”

and

“The large, velocity-dependent scatter of the Hall angle is consistent with the plastic flow regime in which moving and pinned skyrmions coexist.”

to

“The large, non-zero, velocity-dependent scatter of the Hall angle is consistent with the flow regime in which moving and pinned skyrmions coexist.”

(3) Just some points regarding the background of the system. The first predictions for a drive dependent skyrmion Hall angle were presented in 2015, where it was argued to be due to a pinning effect:

Collective Transport Properties of Driven Skyrmions with Random Disorder

C. Reichhardt, D. Ray, and C. J. Olson Reichhardt

Phys. Rev. Lett. **114, 217202 (2015)**

Quantized transport for a skyrmion moving on a two-dimensional periodic substrate

C. Reichhardt, D. Ray, and C. J. Olson Reichhardt

Phys. Rev. B **91, 104426 (2015) - Published 26 March 2015**

This was followed up by additional work

Fluctuations and noise signatures of driven magnetic skyrmions

Sebastián A. Díaz, C. J. O. Reichhardt, Daniel P. Arovas, Avadh Saxena, and C. Reichhardt
Phys. Rev. B 96, 085106 (2017)

C. Reichhardt and C. J. O. Reichhardt, Noise fluctuations and drive dependence of the skyrmion Hall effect in disordered systems, New J. Phys. 18, 095005 (2016).

"Thermal creep and the skyrmion Hall angle in driven skyrmion crystals",

C. Reichhardt and C.J.O. Reichhardt
J. Phys.: Condens. Matter 31 07LT01 (2019)

Also regarding the "controversy", the only work that stated that it was the skyrmion shape or current effect that caused the skyrmion Hall angle to change was:

K. Litzius, I. Lemesh, B. Krüger, P. Bassirian, L. Caretta, K. Richter, F. Büttner, K. Sato, O. A. Tretiakov, J. Förster, R. M. Reeve, M. Weigand, I. Bykova, H. Stoll, G. Schütz, G. S. D. Beach, and M. Kläui, Skyrmion Hall effect revealed by direct time-resolved X-ray microscopy, Nat. Phys. 13, 170 (2017).

However, this work has no theory or calculation to support this claim; it was simply stated. Also there are now several continuum based simulations that see no change in the skyrmion Hall angle when pinning is absent.

Thank you very much for the provided background. We have taken the comments on board and made the following changes to the manuscript, including the addition of several of these references.

"Therefore, one expects a velocity independent θ_{sky} of 10° . This expression depends only on the skyrmion geometry and the damping constant, and so does not predict any dependence of θ_{sky} on the driving force. However, experiments so far have demonstrated that multilayer systems show an unexpected current-density, i.e. driving force dependence of the skyrmion Hall angle^{20,21}. In particular, a linear dependence of the skyrmion Hall angle on velocity has been reported²⁰⁻²². "

Now reads

"Therefore, one expects a velocity independent θ_{sky} of 10° . This expression depends only on the skyrmion geometry and the damping constant, and so does not predict any dependence of θ_{sky} on the driving force. However, the above assumes a perfectly clean system. When defects are introduced deviations from the ideal case are predicted²¹⁻²⁴. Experimentally, a current density, i.e., driving force dependence of the skyrmion Hall angle, was reported in multilayer systems^{20,25,26}. In particular, a linear dependence of the skyrmion Hall angle on velocity was observed^{20,25,26}"

The paper cited in Ref

Lin, S., Reichhardt, C., Batista, C. D. & Saxena, A. Particle model for skyrmions in metallic chiral magnets : Dynamics , pinning , and creep. Phys. Rev. B87, 214419 (2013)

talks about pinning but not the skyrmion Hall effect.

Thank you very much. We have amended the following in the manuscript to improve clarity

"While it is clear that nanoscale variations in the magnetic parameters of devices lead to skyrmion deformation and a wide range of stable diameters^{23,24}, their influence on motion and the skyrmion Hall angle remains an active field of debate^{20,22,23,25-27}. On the one hand, skyrmion Hall angle

deviations are attributed to dynamic deformation of the skyrmion during the motion²¹ and on the other hand deviations are attributed to magnetic grains within the material^{22,23,27} and defects^{25,26}”

now reads

“While it is clear that nanoscale variations in the magnetic parameters of devices lead to skyrmion deformation and a wide range of stable diameters^{27,28}, the influence of defects on motion²⁹ and the skyrmion Hall angle remains an active field of debate^{20,23,26,27,30,31}. On the one hand, skyrmion Hall angle deviations are attributed to dynamic deformation of the skyrmion during the motion²⁵ and on the other hand deviations are attributed to magnetic grains within the material^{26,27,30} and defects²⁰⁻²⁴.”

REVIEWERS' COMMENTS:

Reviewer #1 (Remarks to the Author):

The revised MS addresses all my previous comments in an adequate manner. I recommend to publish as is.

Reviewer #2 (Remarks to the Author):

In my opinion, authors have answered all my previous comments and comments from other referees successfully. I expect this manuscript could excite more intriguing research at this topic. This manuscript can now be published in Nature Com.

Reviewer #3 (Remarks to the Author):

The authors have addressed the points I have raised and have also made changes to address the points of the other two referees. I think with these changes the paper has been improved and can now be published.

Re-Submission of NCOMMS-19-25559A

"Diameter-independent skyrmion Hall angle observed in chiral magnetic multilayers"

We thank the reviewers for their time, their positive reviews and their support for our manuscript.

Reviewer #1 (Remarks to the Author):

The revised MS addresses all my previous comments in an adequate manner.
I recommend to publish as is.

Thank you very much.

Reviewer #2 (Remarks to the Author):

In my opinion, authors have answered all my previous comments and comments from other referees successfully. I expect this manuscript could excite more intriguing research at this topic. This manuscript can now be published in Nature Com.

Thank you very much we agree it should excite more intriguing research.

Reviewer #3 (Remarks to the Author):

The authors have addressed the points I have raised and have also made changes to address the points of the other two referees. I think with these changes the paper has been improved and can now be published.

Thank you very much for your comment.